# Ultra-High Frequency UltraSound (UHFUS) Assessment of Barrier Function in Moderate-to-Severe Atopic Dermatitis during Dupilumab Treatment

**DOI:** 10.3390/diagnostics13172721

**Published:** 2023-08-22

**Authors:** Valentina Dini, Michela Iannone, Alessandra Michelucci, Flavia Manzo Margiotta, Giammarco Granieri, Giorgia Salvia, Teresa Oranges, Agata Janowska, Riccardo Morganti, Marco Romanelli

**Affiliations:** 1Department of Dermatology, University of Pisa, 56126 Pisa, Italy; valentina.dini@unipi.it (V.D.); drmichelaiannone@gmail.com (M.I.); manzomargiottaflavia@gmail.com (F.M.M.); giammarcogranieri@gmail.com (G.G.); giorgia.salvia2@gmail.com (G.S.); dottoressajanowska@gmail.com (A.J.); marco.romanelli@unipi.it (M.R.); 2Unit of Dermatology, Department of Pediatrics, IRCCS Meyer Children’s Hospital, 50139 Florence, Italy; teresa.oranges@gmail.com; 3Statistical Support to Clinical Trials Department, University of Pisa, 56126 Pisa, Italy; r.morganti@ao-pisa.toscana.it

**Keywords:** atopic dermatitis, ultra-high frequency ultrasound, SLEB, dupilumab, corneometry

## Abstract

Atopic dermatitis (AD) is a chronic multifactorial inflammatory disease characterized by intense itching and inflammatory eczematous lesions. Biological disease-modifying drugs, such as dupilumab are recommended for patients with moderate-to-severe AD, refractory to systemic immunosuppressive therapies. Disease monitoring is performed by clinical scores. Since 1970, however, the use of ultrasound and particularly high-frequency ultrasound (HFUS), has identified alterations in dermal echogenicity, called the subepidermal low-echogenic band (SLEB), that correlates with disease severity and response to treatment. We enrolled 18 patients with moderate-to-severe AD, divided into two groups: twelve patients in the dupilumab treatment (Group A) and six patients in standard treatment, from February 2019 to November 2019. We performed ultra-high frequency ultrasound (UHFUS) evaluation of lesional and non-lesional skin, focusing on SLEB average thicknesses measurement, epidermal thickness, and vascular signal in correlation with objective disease scores (EASI, IGA), patient’s reported scores (Sleep Quality NRS and Itch NRS), and TEWL and corneometry at baseline (T0), after 1 month (T1) and 2 months (T2). The SLEB average thickness measurement, vascular signal, and epidermal thickness showed a statistically significant reduction in lesional skin of the biological treatment group and no significant reduction in non-lesional skin in both groups. In the lesional skin of the standard treatment group, only epidermal thickness showed a statistically significant reduction. Our study demonstrates that SLEB measurement, vascular signals, and epidermal thickness could be used as objective parameters in monitoring the AD treatment response, while the presence of SLEB in non-lesional skin could be used as a marker of subclinical inflammation and could predict development of clinical lesions, suggesting a pro-active therapy. Further follow-up and research are needed to clarify the association of SLEB decrease/disappearance with a reduction of flares/prolongment of the disease remission time.

## 1. Introduction

Atopic dermatitis (AD) is a chronic, relapsing cutaneous inflammatory disease characterized by the interaction of genetic factors, environmental factors, immune abnormalities, and comorbidities [1]. In developed countries, AD affects one-fifth of the population and has been identified as the most common inflammatory skin disease. In particular, it has been demonstrated that AD affects approximately 5 to over 20 percent of children worldwide, with higher rates in Africa, Oceania, and the Asia–Pacific region rather than in Northern/Eastern Europe [2,3]. The onset of the disease frequently occurs before the age of five years, with early onset in the first six months of life that seems to be associated with severe disease and with a slight preponderance of prevalence in female children [4]. Clinical features include intense itching and inflammatory eczematous lesions [5]. Diagnosis of AD is based on a typical clinical picture without diagnostic markers; practical aspects of management, such as duration of treatment, criteria for switching, and determination of severity, are insufficient. Disease severity is determined using scales such as the Atopic Dermatitis Score Index (SCORAD), the Eczema Area and Severity Index (EASI), or the Investigator Global Assessment (IGA), which measures the degree of erythema, induration/papule/edema, abrasions, and lichenification according to the physician and the degree of itching and sleep disturbance according to the patient [6].

The chronic nature of the disease means that long-term treatment strategies are necessary, which entails high global healthcare costs and significant psychosocial implications for patients and their relatives [7]. In particular, AD imposes a significant economic burden on healthcare systems, including direct medical costs of diagnosis and treatment, along with indirect costs related to productivity loss and intangible costs, such as reduced quality of life and psychological distress [8]. Efforts to address the economic burden of AD should focus on early diagnosis, effective treatments, patient education, and supportive care. Investing in the management of AD can lead to better patient outcomes, improved quality of life, and potentially reduced long-term healthcare costs.

The optimal management of AD is based on a multimodal approach that involves, first, correct patient education on the importance of skin hydration, avoidance of trigger factors, and implementation of the skin barrier function, highlighting that pharmacologic treatment of skin inflammation represents a second step that cannot ignore the first one. Although there is no definitive cure, the disease can be controlled with appropriate treatment. Current therapy recommendations include topical moisturizers to restore epidermal barrier function, topical corticosteroids (TCS) to control the acute outbreak, topical calcineurin inhibitors (TCI) for sensitive skin areas and long-term use, and phototherapy (preferably UVB 311 nm or UVA1) as adjuvant therapy [9]. In severe refractory cases, systemic immunosuppressive treatments such as cyclosporine, methotrexate, azathioprine, and mycophenolic acid are recommended. Biologic disease modifiers such as dupilumab are recommended “for patients with moderate to severe AD for whom local treatment is inadequate and other systemic treatments are undesirable” [10]. In particular, dupilumab is a human monoclonal antibody able to inhibit IL-4 and IL-13 signaling through a link to the IL-4R-alpha and IL-13R-alpha-1 subunits of the receptor [11]. The blocking of IL-4 is responsible for under-stimulating B-cell differentiation and IgE production [12], leading to the regulation of receptor signaling downstream of the JAKSTAT pathway, with the subsequent activation of tyrosine kinases 2 and Januskinase (JAK) 1/2, resulting in gene expression regulation [13]. In particular, JAK-STAT activation is able to down-regulate skin-barrier proteins and interfere with keratinocyte differentiation, defeating the decrease of skin barrier function caused by the down-regulation of the filaggrin protein [14].

Moreover, AD shows a chronic and relapsing course, with several flares of the disease, which usually undergo spontaneous remission in cases of mild disease, while moderate to severe dermatitis rarely clears without treatment. The monitoring of the flares is mainly clinical, but several non-invasive methods have been proposed for objective evaluation of the disease, independently from the chosen therapy. The first studies on non-invasive assessment of skin biophysical parameters were published as early as the late 1990s. Transepidermal water loss (TEWL) and corneometry were the most evaluated parameters [15]. TEWL is a critical parameter that measures the amount of water lost through the epidermis to the external environment. A healthy skin barrier effectively retains moisture, preventing excessive water loss and maintaining skin hydration. However, in AD, the disrupted barrier function results in elevated TEWL levels, contributing to the hallmark symptoms of dry and itchy skin [16]. Excessive TEWL exacerbates the inflammatory response in AD by triggering a cascade of immune reactions. It also weakens the skin’s ability to heal, perpetuating skin lesions and further compromising the barrier function. Therefore, understanding and managing TEWL play a crucial role in the treatment and management of AD. On the other side, corneometry is a non-invasive technique used to assess the skin’s hydration levels, primarily by measuring the electrical capacitance of the skin surface. The principle behind corneometry is based on the fact that water is a potent electrical conductor, and changes in skin hydration affect its electrical properties; therefore, by measuring the skin’s electrical capacitance, corneometry provides a quantitative evaluation of its hydration status [17]. Corneometry has become an important tool in the evaluation and management of AD, since it allows dermatologists to objectively measure and monitor the hydration status of the skin, providing valuable insights into disease severity and treatment response [18]. In patients with active AD flares, corneometry often reveals significantly lower hydration values, indicating decreased skin moisture content. Monitoring hydration levels during treatment can help clinicians assess the efficacy of topical moisturizers and systemic therapies, such as immunomodulators and corticosteroids [19]. For all these reasons, corneometry has been established as a valid tool also for the identification of individuals at risk of developing AD, as early changes in skin hydration may precede the appearance of clinical symptoms.

Another non-invasive diagnostic method was explored in 1979, when Alexander and Miller Skin introduced ultrasound into dermatology to measure skin thickness. Since the development of high-frequency ultrasonography (HF-US) and its application in the non-invasive monitoring of inflammatory skin diseases, significant advancements have been made in understanding and diagnosing conditions such as psoriasis, eczema, and atopic dermatitis (AD). One prominent feature observed in these inflammatory skin diseases is the presence of a subepidermal low-echogenic band (SLEB), which indicates skin edema and infiltration by inflammatory cells, resulting in increased distance between collagen fibers [20]. Research has demonstrated a correlation between the thickness of the SLEB and the severity of AD, making it a valuable objective parameter for monitoring the disease’s course [21]. Moreover, studies using HF-US have revealed that non-lesional AD skin exhibits a thinner SLEB compared to healthy controls, suggesting the presence of subclinical eczematous lesions [22,23]. These findings underscore the potential of HF-US as a non-invasive diagnostic tool and its utility in monitoring AD patients over time. Despite the promising developments in using HF-US for disease monitoring, there is a notable gap in the literature regarding a multimodal assessment of changes in the skin barrier function following various systemic and/or biological therapies used for managing AD. This area of research remains relatively unexplored, and there is a need for more comprehensive studies to investigate the impact of different treatment modalities on the skin barrier function in AD patients. A multimodal assessment would entail integrating data from HF-US with other diagnostic techniques, such as dermatological assessments, biopsies, and measurements of biomarkers related to skin barrier function. By combining these approaches, researchers and clinicians could gain a more comprehensive understanding of how specific therapies affect the skin barrier in AD patients and whether these changes correlate with treatment response and disease outcome. Such studies would not only advance our knowledge of the underlying mechanisms of AD but also provide valuable insights for tailoring personalized treatment plans for patients. Additionally, understanding the impact of different therapies on the skin barrier could lead to the development of novel treatment strategies targeting the restoration of barrier integrity, ultimately improving the management and quality of life for AD patients.

Since then, new applications of high-frequency ultrasonography (HF-US) have been developed, including non-invasive monitoring of inflammatory skin diseases. In inflammatory skin diseases (psoriasis, eczema, AD), a subepidermal echogenic or hypoechogenic band (SLEB) is observed, and the average skin echogenicity is reduced; SLEB shows skin edema and infiltration by inflammatory cells with increased distance between collagen fibers [20]. It has been demonstrated that SLEB thickness correlates with the severity of AD and can be used as an objective parameter to monitor the course of the disease [21]. Moreover, HF-US studies reported a thinner SLEB in non-lesional AD skin compared to higher cutaneous US in controls, with a thinner SLEB that may generally indicate a subclinical eczematous lesion [22,23]. Even if non-invasive diagnostic has then been shown to represent an opportunity for the follow-up of AD patients, curiously, there are still no studies in the literature that perform a multimodal assessment of the changes in the barrier function following the various systemic and/or biological therapies available for the management of AD.

## 2. Materials and Methods

A prospective study was conducted from February 2019 to November 2019 at the Department of Dermatology of the University of Pisa, Pisa, Italy, on AD patients who were followed with non-invasive measurements of affected and unaffected skin. The study was conducted under the 1964 Declaration of Helsinki and all subsequent amendments, and all patients provided informed consent. Eligible subjects were patients aged ≥ 18 years affected by the classic phenotype of moderate to severe AD; subjects with erythrodermic or prurigo nodularis forms were excluded, as well as patients who were lost to follow-up, or who had any biologic treatment interruption, or the switching from biologic therapy to another biologic therapy. The patients were divided into two groups: Group A included those patients in the biological treatment, while Group B consisted of patients treated with standard treatment. For Group A, we included all patients who received Dupilumab 300 mg administered subcutaneous according to the dosing schedule at week 0, 4, and then every 2 weeks. Clinical and instrumental evaluations were performed at baseline (T0), after 1 month (T1), and 2 months (T2). At each visit, an objective examination based on classic disease scores (EASI, IGA) and patient-reported scores (NRS sleep quality, NRS itching) was performed. For each patient, instrumental examinations were performed on the lesional skin of the antecubital cavity and on the non-lesional skin of the contralateral antecubital cavity not affected by eczematous lesions. TEWL and corneometry were measured by Dermalab^®^ COMBO (Cortex Technology, Hadsund, Danmark) according to EEMCO Group guidelines (European Expert Group on Efficacy Measurement of Cosmetics and Other Topical Products). Each assessment was always conducted under identical environmental conditions after 15–30 min of acclimation; TEWL was expressed in international units (g/m^2^/h) and ranged from 0 to 250 g/m^2^/h (normal values were 0–25 g/m^2^/h). Corneometry was expressed in microsimens (μS) and ranged from 0 to 9999 μS. Depending on the instrument settings, eight measurements were taken and the average value automatically generated by the software was taken into account. To further investigate skin barrier function and inflammation, an ultra-high frequency ultrasound (UHFUS) was performed on a VEVO MD^®^ (FUJIFILM VisualSonics, Toronto, Ontario, Canada) using a 70 MHz ultrasound probe with a frequency range of 29–71 MHz, axial resolution of 30 μm, lateral resolution of 65 μm, depth of 10 mm and maximum width of 9.7 mm (B MODE 31, C-MODE speed 1.9 cm/s). Longitudinal ultrasound probe video clips, transverse ultrasound probe video clips, and color Doppler studies (both longitudinal and transverse) at a standard di 1.9 cm/s were performed for each skin area under examination. Ultrasonography was performed using a sufficient amount of ultrasound gel between the probe and the skin. The probe was held perpendicular to the skin with minimal pressure and moved manually. Two dermatologists trained in skin ultrasonography practiced ultrasonography and imaging. For each ultrasound video clip, SLEB, epidermis and dermis thickness were measured using RadiAnt DICOM Viewer^®^ software (Medixant, v.5.0.1.21910) and displayed in millimeters. Three trained operators chose the three most significative frames in each image, both longitudinal and transverse, and they practiced three measurements of each parameter. The final value was calculated as the average of the three measurements for both images.

The level of vascularization, evaluated both in longitudinal and transverse axes, was assessed using color Doppler imaging at a standard rate of 1.9 cm/s. For each image, the vascularization level was determined in a 2.0 mm depth window. The same settings were used for images of healthy skin. Color Doppler signal was assessed in quantitative scores from 0 to 3: 0—none; 1—weak (physiological); 2—moderate; 3—strong. The final numerical score was calculated by average longitudinal and transverse image scores. All categorical data were described by absolute and relative (%) frequency, and continuous data by mean and standard deviation. To evaluate the differences between groups in terms of clinical scores and non-invasive measurements, ANOVA for repeated measures was performed. Significance was fixed at 0.05, and all analyses were carried out with SPSS v.28 technology.

## 3. Results

A total of 18 subjects, 11/18 (61%) males and 7/18 (39%) females, with a mean BMI of 24.6 kg/m^2^ affected by eczematous lesions of the classic clinical form of AD were included in the study. Age ranged from 20 to 77 years, the mean age was 46 years and the standard deviation was 18. 10/18 patients (55%) presented phototype II and 8/18 (45%) phototype III. Group A included 12/18 (66.7%) patients who were treated with dupilumab, while Group B included 6/18 (33.3%) patients who were treated as follows: 1/12 (8.3%) with cyclosporine, 1/12 (8.3%) with ultraviolet phototherapy, 3/12 (25%) with TCI, and 1/12 (8.3%) with TCS. The mean values and standard deviations of clinical parameters, ultrasound parameters, TEWL, and corneometry at T0, T1, and T2 are reported in Table 1 for patients who received biological treatment (Group A) and in Table 2 for patients who received standard treatment (Group B). The two groups displayed some clinical differences at baseline, since Group A moved from a mean EASI of 28.425 (sd: 12.339) and IGA of 3.250 (sd: 0.452), while Group B from a mean EASI of 12.050 (sd: 7.584) and IGA of 2.167 (sd: 0.753). Also the quality of life of patients of Group A was more compromised, with a mean DLQI of 12.750 (sd: 6.426), itch-NRS of 8.5 (sd: 1.382), and sleep-NRS of 5.750 (sd: 3.019); on the other side, Group B showed a mean DLQI of 9.833 (sd: 6.494), an itch-NRS of 6.667 (sd: 1.033), and sleep-NRS of 3.333 (sd: 2.944). Focusing on UHFUS measurements, Group A presented an initial lesional skin SLEB of 0.291 mm(sd: 0.126), non-lesional skin SLEB of 0.053 mm (sd: 0.040), a lesional skin epidermic thickness of 0.177 mm (sd: 0.030), a non-lesional skin epidermic thickness of 0.146 mm (sd: 0.027), a lesional skin dermis thickness of 1.749 mm (sd: 0.480), a non-lesional skin dermis thickness of 1.559 mm (sd: 0.451), a lesional skin vascularization of 2.292 (sd: 0.480), and a non-lesional skin vascularization of 0.750 (sd: 0.584). Results from Group B were similar, with some lower values as well as the clinical ones. Indeed, we registered an initial lesional skin SLEB of 0.195 mm (sd: 0.085), non-lesional skin SLEB of 0.038 mm (sd: 0.049), a lesional skin epidermic thickness of 0.171 mm (sd: 0.055), a non-lesional skin epidermic thickness of 0.138 mm (sd: 0.044), a lesional skin dermis thickness of 1.544 mm (sd: 0.273), a non-lesional skin dermis thickness of 1.311 mm (sd: 0.345), a lesional skin vascularization of 2.167 (sd: 0.516), and a non-lesional skin vascularization of 0.583 (sd: 0.492). TEWL measurements pointed out a mean value of lesional skin TEWL of 40.833 (sd: 22.904) in Group A and of 40.983 (sd: 21.505) in Group B, while non-lesional skin TEWL showed a mean value of 16.067 (sd: 11.271) in Group A and of 28,417 (sd: 22.668) in Group B. To conclude baseline measurements, mean lesional skin corneometry of Group A was 131.00 (sd: 68,621) vs. 248.33 (sd: 257.498) in Group B; furthermore, the mean non-lesional skin corneometry of Group A was 107,333 (sd: 39,919) vs. 194.50 (sd: 1496.020) in Group B.

For Group A, a significant reduction in the objective clinical parameters of EASI, IGA (*p*-value < 0.001) was found, as well as an improvement in the quality of life measured through DLQI (*p*-value 0.001) and both Itch NRS (*p*-value < 0.001) and Sleep-NRS (*p*-value = 0.002). Similar results were obtained in Group B, with a slighter reduction of EASI (*p*-value = 0.02), IGA (*p*-value = 0.063), DLQI (*p*-value = 0.083), and Itch NRS (*p*-value = 0.033), while Sleep-NRS did not reach a statistically significant improvement. Focusing on non-invasive measurements, none of the two groups presented a significant reduction in TEWL and corneometry. Conversely, UHFUS evaluations pointed out differences in Group A in terms of mean lesional skin SLEB (*p*-value = 0.001), lesional skin epidermic thickness (*p*-value = 0.002), and lesional skin vascularization measured through Doppler (*p*-value < 0.001). In the standard treatment Group B, only the epidermal thickness of lesional skin showed a statistically significant decrease (*p*-value = 0.043). Non-lesional skin parameters did not display any significant modifications in either Group A or in Group B.

## 4. Discussion

Patients affected by moderate to severe forms of AD often require systemic treatment to achieve adequate disease control independently from a correct and optimal topical therapy [24,25]. On the contrary, a recent meta-analysis demonstrated that mild to moderate forms of AD can benefit from intermittent therapy with moderate- to high-potency TCS or TCI, which was able to reduce the risk of flares after the disease control achieved with continuous use of the agents [26]. In any case, a re-evaluation of patients to exclude concurrent diseases or conditions that may influence the response (e.g., infection, contact dermatitis) is considered a good clinical practice before the start of any systemic therapy, as well continuous monitoring of the clinical response after the choice of the first-line therapy. Dupilumab is a fully human monoclonal antibody approved for the treatment of adults and children with moderate to severe AD not correctly controlled with topical and/or systemic therapies, displaying a favorable safety profile that does not typically require a serum monitoring as non-targeted immunosuppressive agents [27]. Efficacy of dupilumab has been shown in multiple work and real life-experiences, such as a recent network meta-analysis of 74 randomized trials (more than 8000 patients), which declared dupilumab as the most effective treatment in achieving a 75 percent reduction in the EASI (EASI-75) score (risk ratio 3.04, 95% CI 2.51–3.69) and improving the Patient-Oriented Eczema Measure (POEM) score (mean difference 7.3, 95% CI 6.61–8.00) during short-term follow-up when compared with placebo [28]. Moreover, the long-term safety of dupilumab was evaluated in a randomized, double-blind, multicenter trial (LIBERTY AD CHRONOS) on 740 patients [27], who experienced similarly rated adverse events both when treated with dupilumab or receiving placebo plus topical corticosteroids (83 to 88 percent). Beyond its therapeutic efficacy, recent research has shed light on the impact of dupilumab on skin modification, particularly in relation to the skin barrier function and overall cutaneous health. Indeed, research investigating the effects of dupilumab on skin modification at the histological and molecular levels has revealed promising findings. Biopsy studies have shown that dupilumab treatment reduces epidermal thickness, parakeratosis, and spongiosis, all of which are associated with AD severity [29]. Additionally, gene expression analysis has demonstrated that dupilumab shifts the transcriptome of lesional skin toward a more non-pathological phenotype. This means that dupilumab reduces the expression of genes involved in type 2 inflammation and epidermal hyperplasia while increasing the expression of genes associated with epidermal differentiation, barrier function, and lipid metabolism [30]. Even if monitoring of therapeutic response can be frequently performed through a clinical evaluation and correct dialogue with the patient, the study of residual inflammation and skin modification, which can lead to new flares, is not easily performed during the clinical practice. From a molecular point of view, non-lesional skin of AD patients does not have the structure of normal skin from healthy controls [31], with deep differences in terms of the general skin barrier function, qualitative and quantitative changes in dendritic cell populations, and lymphocytic infiltration. In particular, clinical severity seems to be related to the mean number of infiltrating dendritic cells as well as disease recurrence was demonstrated to be driven by the repopulation of cutaneous inflammatory dendritic cells [32]. All this molecular evidence suggests that there is still limited knowledge on managing subclinical inflammation, and few studies have been performed on managing clinical responses through non-invasive dermatological instruments. UHFUS allows the clinician to obtain real-time images with the possibility of performing measurements of physiological and pathological aspects of the skin. According to data presented in the literature, dermal echogenicity is influenced by collagen fiber location and water content [33,34,35]. Damage to collagen fibers leads to a reduction in the echogenicity of the dermis, which is visible in different inflammatory diseases due to swelling of the skin and inflammatory cell infiltration. Although SLEB is not a specific parameter for any skin disease, its changes over time have important prognostic value, especially for patients with chronic skin diseases such as psoriasis and AD. SLEB thickness in AD correlates with histopathological aspects such as epidermal hyperplasia and hyperkeratosis, levels of parakeratosis and spongiosis, and inflammatory cell infiltration [36,37]. Moving from these assumptions, it is not surprising that even in our work, UHFUS measurements have revealed a parallel reduction in skin thickness and inflammation in response to dupilumab therapy. The most valuable result of our study is that SLEB scores, vascular signals, and epidermal thickness were much more significantly reduced in patients receiving biological treatment compared to patients with standard treatment. The results obtained from this study support the possibility of considering SLEB as an objective parameter for monitoring treatment efficacy in AD [20]. In particular, as can be seen from Figure 1, there is a parallel reduction in SLEB, vascular signs, and epidermal thickness in the lesional skin of both treated groups. These results can be explained by the ability of biological drugs to gradually shift lesion transcriptomes toward a non-pathological phenotype, reducing the expression of genes involved in type 2 inflammation and epidermal hyperplasia and increasing the expression of epidermal differentiation, barrier, and lipid metabolism genes [29]. According to the literature, SLEB at baseline was measurable on the affected skin in 100% of patients with AD [21,22,23]. In 72.2% of patients with AD, SLEB could also be detected on non-lesional skin. These data are significantly higher than those reported in the literature and can be explained by considering that ultrasound was performed using a 70 MHz probe (UHFUS), with an axial resolution of 30 μm and transverse resolution of 65 μm, which enable better skin resolution [22,23]. A statistically significant decrease in SLEB correlates with AD severity and may be an indicator of treatment effectiveness. The presence of a hypoechogenic band also in the perilesional skin of AD subjects would support the idea that there are subclinical eczematous reactions predictive of disease reactivation [23]. Particularly, the persistence of subclinical inflammation could identify patients who need to change or intensify therapy, as previously indicated in the literature [38,39]. In the lesional skin of patients with AD, a moderately-intensive vascular signal was detected in 100% of cases at baseline and not detected at T1 and T2. Physiological patterns of vascularization at baseline and follow-up visits were detected in non-lesional skin. A statistically significant decrease in vascular signal from baseline to T1 and T2 represents a useful marker of clinical inflammation. Data on vascular signals in AD affected and non-affected skin has been poorly evaluated in the literature. Conversely, vascular signals have been used to assess angiogenesis and malignant and metastatic potential in pigmented skin lesions and as a marker of clinical inflammation in psoriasis and hidradenitis suppurativa [40]. In AD, the vascular signal can be a useful marker of clinical inflammation, but unlike SLEB, it cannot be considered a marker of subclinical inflammation. Studies have shown that angiogenesis is increased in the lesional skin of individuals with AD, and the increased blood vessel density and vascular permeability contribute to skin redness and edema observed in AD flare-ups [41]. Additionally, the newly formed blood vessels facilitate the recruitment of immune cells to the affected skin, amplifying the inflammatory response even through vascular endothelial growth factor (VEGF), one of the key angiogenic factors implicated in AD and whose levels are Increased in AD skin lesions [42]. In Figure 2, there is a representation of the clinical and UHFUS evolution of a patient treated with dupilumab during the different time sets of our study, with a clear reduction of the vascular signal measured through the Doppler function.

## 5. Conclusions

Our study displays a few limitations. The first one is represented by the differences in some clinical parameters at T0 between the two selected groups. However, the differences in the two groups can be easily explained by taking into account that Group A included patient candidates for the biologic therapy, which in Italy can be prescribed in adults patients only in the presence of a high burden of the disease measured through the classical clinical score of EASI that, consequently, leads to differences even in terms of IGA and quality of life indexes [43,44]. Another limitation is represented by the sample size. We expect that this pilot study can be expanded to achieve a higher level of statistical significance. It would also be interesting to include patients with different AD severity to objectify the correlation between clinical severity assessment and ultrasound parameters. Our study shows that SLEB measurements, vascular signals, and epidermal thickness can be used as objective parameters to monitor treatment efficacy in AD. To date, treatment monitoring has always been based on clinical remission of active lesions, but the presence of SLEB in non-lesional skin can be used as a marker of subclinical inflammation and can predict the development of clinical lesions, requiring active treatment. In conclusion, SLEB is a non-invasive, safe, and reproducible parameter, and it seems appropriate to combine ultrasound features of lesional and non-lesional skin with clinical scores to assess disease severity. Further investigations are needed to clarify the relationship between a decrease or disappearance of SLEB and a reduction in flares and prolongment of disease remission time. Combining UHFUS with clinical scores allows for a comprehensive evaluation of disease activity, guiding treatment decisions for better patient outcomes. Although this pilot study shows promising results, further research and larger-scale studies are necessary to validate the utility of UHFUS in routine clinical practice. The integration of advanced imaging techniques like UHFUS holds great potential to revolutionize AD management, leading to more effective and personalized treatment approaches for patients affected by this challenging skin condition.

## Figures and Tables

**Figure 1 diagnostics-13-02721-f001:**
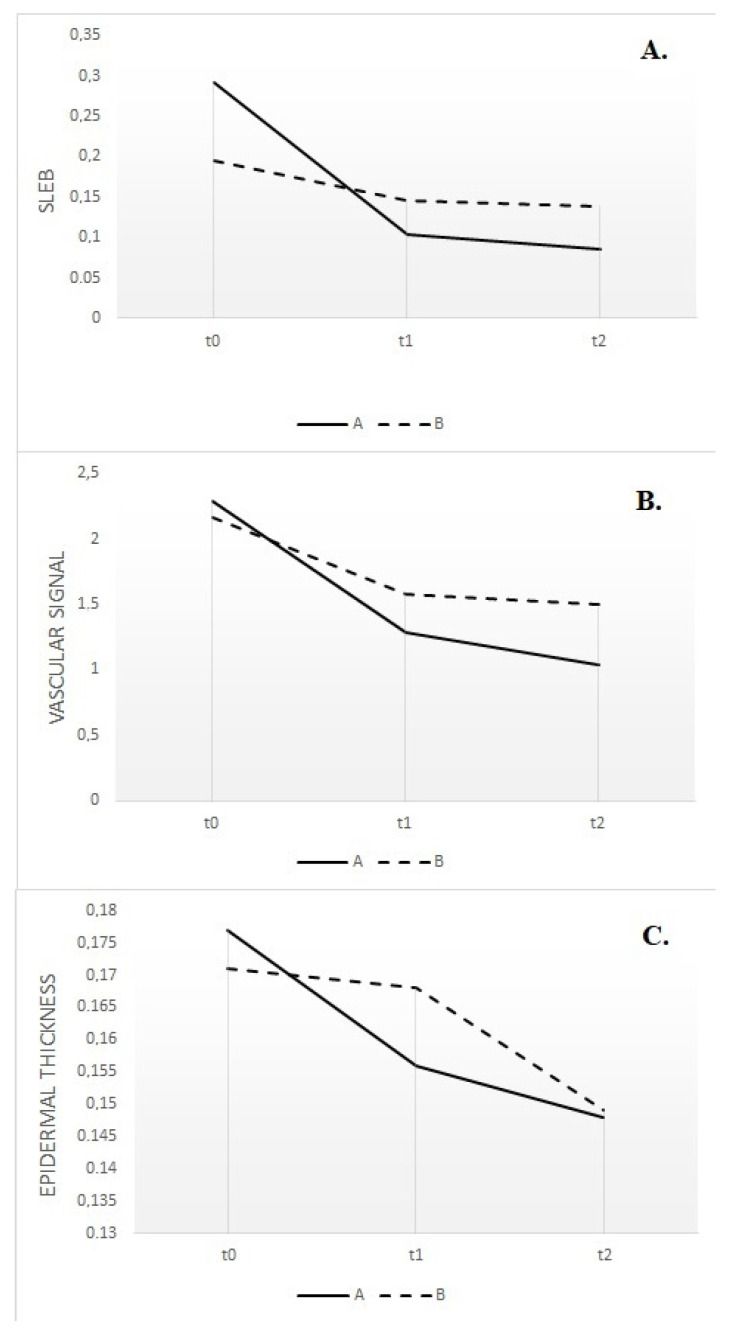
Subepidermal low-echogenic band (SLEB) reduction in lesional skin of the two groups, evaluated at baseline (t0), after 1 month (t1) and 2 months (t2) (**A**); the vascular signal in lesional skin of the two groups, evaluated at baseline (t0), after 1 month (t1) and 2 months (t2) (**B**); epidermal thickness in lesional skin of the two groups, evaluated at baseline (t0), after 1 month (t1) and 2 months (t2) (**C**).

**Figure 2 diagnostics-13-02721-f002:**
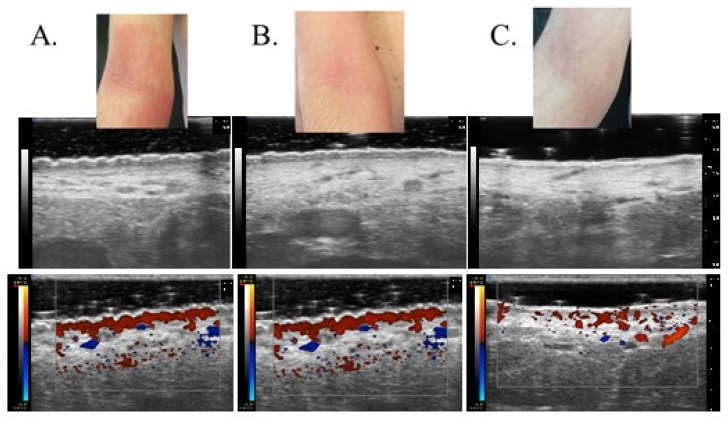
Clinical and UHFUS examination of a patient treated with dupilumab evaluated at baseline (t0) (**A**), after 1 month (t1) (**B**) and 2 months (t2) (**C**).

**Table 1 diagnostics-13-02721-t001:** Mean values (sd) for patients in biological treatment (Group A), evaluated at baseline (t0), after 1 month (t1) and 2 months (t2).

Time Line	t0		t1		t2		
PARAMETERS	Mean	SD	Mean	SD	Mean	SD	*p*-Value
EASI	28.425	12.339	9.875	9.360	8.033	8.815	<0.001
IGA	3.250	0.452	1.833	0.718	1.750	0.866	<0.001
DLQI	12,750	6.426	3.667	3.284	3.417	3.942	0.001
Sleep-NRS	5.750	3.019	1.167	1.946	2.417	2.778	0.002
Itch-NRS	8.500	1.382	3.250	2.340	3.333	2.309	<0.001
Lesional skin SLEB (mm)	0.291	0.126	0.103	0.072	0.085	0.100	<0.001
Non-lesional skin SLEB (mm)	0.053	0.040	0.057	0.080	0.030	0.035	ns
Lesional skin epidermic thickness (mm)	0.177	0.030	0.156	0.037	0.148	0.028	0.002
Non-lesional skin epidermic thickness (mm)	0.146	0.027	0.142	0.023	0.138	0.020	ns
Lesional skin dermis thickness (mm)	1.749	0.480	1.656	0.690	1.607	0.678	ns
Non-lesional skin dermis thickness (mm)	1.559	0.451	1.650	0.567	1.589	0.527	ns
Lesional skin vascularization	2.292	0.450	1.292	0.450	1.042	0.396	<0.001
Non-lesional skin vascularization	0.750	0.584	0.625	0.483	0.625	0.483	ns
Lesion skin TEWL	40.833	22,904	29,250	12,624	28,150	12,697	ns
Non-lesional skin TEWL	16,067	11,271	21,583	15,967	14,392	9.104	ns
Lesion skin corneometry	131,000	68,621	132,667	81,615	103,833	37,365	ns
Non-lesional skin corneometry	107,333	39,919	119,667	83,049	101,167	33,526	ns

Eczema Area and Severity Index (EASI), Investigator’s Global Assessment (IGA), Dermatology Life Quality Index (DLQI), Numerical Rating Scale (NRS), Subepidermal Low-Echogenic Band (SLEB), Transepidermal water loss (TEWL).

**Table 2 diagnostics-13-02721-t002:** Mean values (sd) for patients in standard treatment (Group B) evaluated at baseline (t0), after 1 month (t1) and 2 months (t2).

Time Line	t0		t1		t2		
PARAMETERS	Mean	SD	Mean	SD	Mean	SD	*p*-Value
EASI	12,050	7.584	6.183	3.945	5.667	5.297	0.020
IGA	2167	0.753	1.667	0.816	1.500	1.049	0.063
DLQI	9833	6.494	4.667	3.011	1.667	1.506	0.083
Sleep-NRS	3333	2.944	1.833	2.483	1.667	2.338	ns
Itch-NRS	6667	1.033	2.833	2.927	3.333	2.338	0.033
Lesional skin SLEB (mm)	0.195	0.085	0.146	0.084	0.139	0.159	ns
Non-lesional skin SLEB (mm)	0.038	0.049	0.050	0.045	0.044	0.040	ns
Lesional skin epidermic thickness (mm)	0.171	0.055	0.168	0.023	0.149	0.034	0.043
Non-lesional skin epidermic thickness (mm)	0.138	0.044	0.127	0.024	0.146	0.019	ns
Lesional skin dermis thickness (mm)	1.544	0.273	1.387	0.200	1.360	0.329	ns
Non-lesional skin dermis thickness (mm)	1.311	0.345	1.341	0.282	1.306	0.374	ns
Lesional skin vascularization	2.167	0.516	1.583	0.585	1.500	0.837	ns
Non-lesional skin vascularization	0.583	0.492	0.667	0.753	0.750	0.274	ns
Lesion skin TEWL	40.983	21,505	28,367	8.641	46,350	25,401	ns
Not-lesional skin TEWL	28,417	22,668	26,000	17,535	41,100	14,846	ns
Lesion skin corneometry	248,333	257,498	117,667	95,406	151,000	62,846	ns
Not-lesional skin corneometry	194,500	146,020	158,000	101,052	166,167	76,583	ns

Eczema Area and Severity Index (EASI), Investigator’s Global Assessment (IGA), Dermatology Life Quality Index (DLQI), Numerical Rating Scale (NRS), subepidermal low-echogenic band (SLEB), transepidermal water loss (TEWL).

## Data Availability

The data presented in this study are available on request from the corresponding author. The data are not publicly available due to privacy reasons.

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
