# Peer review of "Ultra-High Frequency UltraSound (UHFUS) Assessment of Barrier Function in Moderate-to-Severe Atopic Dermatitis during Dupilumab Treatment"

_diagnostics, 2023, doi:10.3390/diagnostics13172721_

Round 1

Reviewer 1 Report

The use of ultra-high frequency ultrasound (UH-FUS) is gaining attention in dermatological disease diagnosis in the last decade, though known for a century. In this work by Dini et al., the authors aim to evaluate the use of UF-FUS in disease prognosis in AD patients receiving either biological (Dupilumab) or conventional (tacrolimus/phototherapy/corticosteroids). In particular, sub-epidermal low echogenic band (SLEB) and vascular signal (acquired as a function of doppler effect), were evaluated for its prognostic potential in both lesional and non-lesional (subclinical inflammation) skin under biological and conventional treatment regimen. Further, the results were also compared with traditional scoring parameters in AD for clinical correlation. Multiple outcomes have emerged in this study. Firstly, the decrease in SLEB thickness and vascular signals are positively correlated with reduced AD severity. Secondly, the decrease in SLEB scores, vascular signal and epidermal thickness is significantly more during the course of biological treatment compared to conventional treatment. Most notably, SLEB was also noted in non-lesional skin of AD patients highlighting its use to diagnose sub-clinical inflammation. However, it does not show a significant change during the treatment course. The authors also acknowledge the limitations of this study such as small sample size and baseline differences between patients chosen for biological and conventional treatment regimen. This manuscript can be accepted provided the following concerns are addressed clearly.

1.     I do not see any unit for SLEB/epidermal thickness. Should not these be in mm?

2.     The title of the manuscript does not clearly outline either the purpose of the manuscript or the key results authors obtained. Though SLEB/UH-FUS takes the center point in this manuscript, I do not see this in title. SLEB is also not there in key words. The title can be improved and may be derived from key conclusions of the manuscript.

3.     More details are required with respect to the clinical manifestation of the patients with respect to their phototype, BMI and if the patients had xerosis etc.

4.     Importantly, SLEB can be influenced by sun exposure and age. Since the subjects are from 20 to 77 years, could there be any influence on patient age on SLEB thickness in different patients? And authors need to properly define the non-lesional skin with respect to how far away from the lesional skin is considered as non-lesional?

5.     The acronym “DA” has been used twice in the manuscript without defining/expanding what does this mean.

Some of the sentences are poorly constructed such as sentences between line nos 75 to 88, line nos 281-282. Sentences may have to be rephrased to convey right meaning

Reviewer 2 Report

This manuscript reports the treatment and evaluation of atopic dermatitis. I recommend publish after minor revision. 

1. The presentation in Introduction should be improved. There is only one paragraph and the logic is confusing. 

2. In Figure 1, the words in the images are too small to be seen. Also, the presentation of y axis is not very clear. The quality of this figure should also be improved.

3. In Figure 2, the scale bar of ultrasound images should be added.

4. Some minor mistakes should be corrected. For example, "DA" should be "AD" in the manuscript.
